EMBO
Molecular Medicine

# Enhancement of red blood cell transfusion compatibility using CRISPR-mediated erythroblast gene editing

Joseph Hawksworth[1,3,†], Timothy J Satchwell[1,2,3,†], Marjolein Meinders[1], Deborah E Daniels[1,2], Fiona Regan[4,5], Nicole M Thornton[6], Marieangela C Wilson[1], Johannes GG Dobbe[7], Geert J Streekstra[7], Kongtana Trakarnsanga[8], Kate J Heesom[1], David J Anstee[1,2,3], Jan Frayne[1,2] & Ashley M Toye[1,2,3,*]

## Abstract

Regular blood transfusion is the cornerstone of care for patients with red blood cell (RBC) disorders such as thalassaemia or sickle-cell disease. With repeated transfusion, alloimmunisation often occurs due to incompatibility at the level of minor blood group antigens. We use CRISPR-mediated genome editing of an immortalised human erythroblast cell line (BEL-A) to generate multiple enucleation competent cell lines deficient in individual blood groups. Edits are combined to generate a single cell line deficient in multiple antigens responsible for the most common transfusion incompatibilities: ABO (Bombay phenotype), Rh ($Rh_{null}$), Kell ($K_O$), Duffy ($Fy_{null}$), GPB (S−s−U−). These cells can be differentiated to generate deformable reticulocytes, illustrating the capacity for coexistence of multiple rare blood group antigen null phenotypes. This study provides the first proof-of-principle demonstration of combinatorial CRISPR-mediated blood group gene editing to generate customisable or multi-compatible RBCs for diagnostic reagents or recipients with complicated matching requirements.

**Keywords** BEL-A; CRISPR; erythroid; transfusion; universal donor
**Subject Category** Haematology

## Introduction

The collection of more than 1.5 million units of blood is required each year to meet the transfusion needs of England alone (MHRA, 2016). For the majority of patients, this need is serviced by the blood donation system which provides hospitals with screened donor blood that is matched for the major blood group antigens: A, B, O and RhD. Donors who are blood group O RhD negative are often described as "universal donors"; however, this popular simplification misrepresents the complexity of RBC surface antigens that influence donor–recipient compatibility with 36 blood group systems and more than 350 different antigens recognised by the International Society of Blood Transfusion (http://www.isbtweb.org/working-parties/red-cell-immunogenetics-and-blood-group-terminology/). Mismatch of any blood group antigen has the potential to cause alloimmunisation (generation of antibodies to non-self RBC antigens). Across all transfusion recipients, this occurs in approximately 2–5% of cases; however, in chronically transfused sickle-cell disease (SCD) patients, RBC alloimmunisation occurs in approximately 30% of cases (Campbell-Lee & Kittles, 2014). Common alloantibodies generated in chronically transfused patients with SCD include those to the C and E antigens of the Rh blood group system (RH), K in the Kell system (KEL), $Fy^a$ and Fy3 in the Duffy system (FY), $Jk^b$ in the Kidd system (JK), and U and S in the MNS blood group system (Rosse et al, 1990; Aygun et al, 2002; Castro et al, 2002). Such incompatibilities often result from the differing prevalence of antigens between SCD recipients of African descent compared to a predominately Caucasian donor base.

Whilst alloimmunisation of chronically transfused patients increases the difficulty in obtaining suitably matched donations, individuals exist with non-pathological but particularly rare naturally occurring blood group phenotypes that also present major challenges for blood transfusion services across the world to match. Examples include individuals with the rare Bombay phenotype [one in 250,000 of the population worldwide (Mallick et al, 2015) and

1 School of Biochemistry, University of Bristol, Bristol, UK
2 NIHR Blood and Transplant Research Unit, University of Bristol, Bristol, UK
3 Bristol Institute for Transfusion Sciences, National Health Service Blood and Transplant (NHSBT), Bristol, UK
4 Imperial College Healthcare NHS Trust, London, UK
5 NHS Blood & Transplant, London, UK
6 International Blood Group Reference Laboratory, National Health Service (NHS) Blood and Transplant, Bristol, UK
7 Department of Biomedical Engineering and Physics, Academic Medical Center, University of Amsterdam, Amsterdam, The Netherlands
8 Department of Biochemistry, Faculty of Medicine Siriraj Hospital, Mahidol University, Bangkok, Thailand
*Corresponding author. Tel: +44 0117 3312111; E-mail: ash.m.toye@bristol.ac.uk
†These authors contributed equally to this work

one in 1,000,000 in Europe (Oriol et al, 2000)] who lack the H antigen—the antigenic determinant of blood group O and the precursor of the A and B antigens—and those who are $Rh_{null}$ [one in every 6,000,000 (Avent & Reid, 2000)] who lack all antigens encoded by both the RHD and RHCE genes. In these, and other rare cases, transfusion with only ABO RhD matched blood will result in adverse effects and therefore donations are required from an individual with the same rare phenotype.

Current efforts to cater for individuals with rare blood types rely on international coordination of rare donor databases and the cryopreservation of donor units including those stored for autologous transfusion (Anstee et al, 1999; Nance, 2009; Nance et al, 2016).The ability to generate RBCs with bespoke phenotypes for individuals with rare blood types together with a "more universal" source of RBCs designed to minimise alloimmunisation in SCD patients and to meet the transfusion needs of existing patients for whom alloimmunisation has reduced the suitable donor pool would have obvious clinical benefits.

The desire to improve RBC compatibility for transfusion is not a new concept. Whilst the successful conversion of blood group types A and B to O has been demonstrated using glycosidases (Zhu et al, 1996; Kruskall et al, 2000; Liu et al, 2007), attempts to more broadly enhance compatibility through antigen masking achieved only limited success (Jeong & Byun, 1996; Armstrong et al, 1997; Scott et al, 1997). More recently, developments in the in vitro culture of erythroid cells have facilitated the laboratory proof-of-principle production of RBCs with severe depletion in the expression of other blood groups using lentiviral expression of shRNAs in haematopoietic stem cells (Bagnis et al, 2009; Cambot et al, 2013). In some cases, these modified cells present the properties of their equivalent null phenotypes as assessed by routine diagnostic haemagglutination tests (Bagnis et al, 2009; Cambot et al, 2013) although remaining residual antigens resulting from incomplete knockdown may still result in alloantibody formation in vivo. An additional obstacle to the use of adult haematopoietic stem cells for the culture of modified RBCs is presented by the finite proliferative capacity of such cells, which currently limits the yield of cultured RBCs to sub-transfusable quantities and the requirement for repeated shRNA transduction between cultures. Alternative approaches which have focused on the derivation of RBCs from more sustainable induced pluripotent stem cells created from existing donors with rare or more universally acceptable RBC phenotypes such as Bombay have been confounded by poor levels of erythroid cell expansion and aberrant or incomplete differentiation (Seifinejad et al, 2010). The generation of immortalised erythroid cell lines provides potential for an unlimited supply of red cells. Gene editing technologies can be utilised in these lines to make modifications which improve transfusion compatibility without the need for repeated editing of haematopoietic stem cells (Kim et al, 2015).

In this study, we use CRISPR-mediated genome editing of a recently developed immortalised human erythroblast cell line BEL-A (Trakarnsanga et al, 2017) to generate stable erythroblast cell lines that can be differentiated to generate functional reticulocytes completely deficient in a variety of individual transfusion-relevant blood groups. By simultaneously expressing multiple guide RNAs (gRNAs) in these cells, we demonstrate the ability to delete multiple blood group genes in erythroblasts and present proof-of-principle generation of red blood cells completely deficient in blood groups encoded by five different genes that encode antigens responsible for the most common transfusion incompatibilities.

## Results

### Target antigen selection

In order to direct the design of a bespoke or customised RBC phenotype that would be able to service existing unmet clinical transfusion needs within the blood transfusion service in England (NHSBT), a survey was conducted to collate instances where rare or problematic transfusion requirements were identified and in which the requirement for matched blood could not be fulfilled or resulted in no remaining store for subsequent patients. This survey, encompassing a combined 15 month period (November 2014–January 2015; April 2015–April 2016), identified 56 patients with rare blood types with alloantibodies against RBC antigens (or with the likelihood of acquisition without specific matching, e.g. in the case of untransfused neonates). Table 1 summarises the data collected in this study; 22 patients had alloantibodies to antigens located on glycophorin B (GPB), U, S or s; 19 patients presented with alloantibodies to at least one antigen within the Rh blood group system; 10 patients possessed alloantibodies to the Duffy blood group [with a further two previously untransfused patients also Fy(a−b−)]; Kell antigen alloantibodies were identified in 10 patients. Eight patients presented with the Bombay or para-Bombay phenotype (a rare phenotype in which a mutation in the FUT1 gene results in the loss of the H antigen and transfusion incompatibility with all ABO blood), and alloantibodies to Lu and Kidd antigens were identified in three patients for each, respectively. In total, 19 patients (the majority of them presenting with SCD) possessed alloantibodies to antigens within more than one blood group system.

This study highlights the major blood groups for which alloimmunisation and transfusion incompatibility is most relevant within the population serviced by NHSBT. Discounting the two McLeod patients [in which absence of XK results in the undesirable trait of acanthocytic erythrocytes (Wimer et al, 1977)], the transfusion requirements of 54 of the 56 patients identified could be serviced by a hypothetical RBC phenotype lacking 7 blood group proteins and 48 of these patients by the removal of just 5. On this basis, we decided to pursue the generation of RBCs completely deficient in surface expression of GPB, Rh, Kell, Duffy and of the Bombay phenotype (able to be received by recipients of blood group A, B, AB, O or Bombay) using CRISPR–Cas9 editing for gene knockout in a recently published immortalised erythroblast cell line: BEL-A (Trakarnsanga et al, 2017).

### Generation of individual blood group knockout cell lines for transfusion therapy and as tools for diagnostics

In addition to acting as a much-needed source of rare blood for transfusion requirements, blood donations by patients with unusual phenotypes play a crucial role in serological testing by blood group reference laboratories. These so-called reagent red cells with established complete deficiency of a given blood group or antigen, for example $Rh_{null}$ or $K_0$, provide essential controls that facilitate the

**Table 1.  Identification of clinical need within NHSBT for rare erythrocyte phenotypes.**

| Blood group system | Patients with alloantibodies | Alloantibodies directed against antigens | Genetic basis of antigens |
|---|---|---|---|
| MNS | 22 | U, S, s | *GYPB* |
| Rh | 19 | D, C, E, c, e, $C^W$, $Hr_0$, $hr^B$, $Hr^B$, MAR | *RHD, RHCE* |
| Duffy | 10 (+2) | $Fy^a$, $Fy^b$, Fy3 | *ACKR1* |
| Kell | 10 | K, k, $Kp^a$ | *KEL* |
| H | 8 | H | *FUT1* |
| Lutheran | 3 | $Lu^a$, $Lu^b$ | *BCAM* |
| Kidd | 3 | $Jk^b$ | *SLC14A1* |

Results of a survey collating instances in which rare or problematic transfusion requirements were identified and in which the requirement for matched blood could not be fulfilled or resulted in no remaining store for subsequent patients. November 2014–January 2015; April 2015–April 2016. Fifty-six patients in total, (+2) indicates untransfused individuals of Fy(a−b−) phenotype yet to develop alloantibodies. Two patients with McLeod syndrome (XK deficiency) are not listed.

**Table 2.  BEL-A genotype.**

| Blood group system | BEL-A predicted phenotype |
|---|---|
| Rh (RH) | D+C−c+E+e+ |
| Duffy (FY) | Fy(a+b−) |
| Kidd (JK) | Jk(a−b+) |
| Kell (KEL) | K−k+ |
| MNS (MNS) | M−N+S−s+ |
| Lutheran (LU) | Lu(a−b+) |
| Diego (DI) | Di(a−b+) |
| Colton (CO) | Co(a+b−) |
| Dombrock (DO) | Do(a+b−) |
| Landsteiner−Wiener (LW) | LW(a+b−) |
| Scianna (SC) | Sc1+Sc2− |

PCR was used to determine the BEL-A genotype for various blood group antigens. Genotyping was performed by the International Blood Group Reference Laboratory.[§]

identification of unknown alloantibodies in patient sera. Such cells, currently curated and stored frozen by selected International Blood Group Reference Laboratories, represent a finite resource, reliant upon the identification of and donations from individuals with often extremely rare naturally occurring mutations and phenotypes. The ability to generate a sustainable library of enucleation competent immortalised erythroblast cell lines completely deficient in expression of a range of individual blood groups of interest would have obvious benefits for serological testing laboratories.

As a first step towards the combinatorial removal of multiple blood groups and for the generation of cell lines with potential therapeutic and diagnostic applications, we sought to generate BEL-A cell lines with individual blood group knockouts useful for these purposes. BEL-A cells were first single cell sorted by fluorescence-activated cell sorting (FACS) and expanded to derive a founder population, expanding cells were lentivirally transduced with a construct containing Cas9 and gRNA targeting the gene of interest as described in Materials and Methods. Following transduction, cells were maintained in culture for at least 1 week to enable turnover of existing protein, labelled with a specific antibody to the extracellular epitope of the blood group protein targeted and then single cell sorted by FACS based on complete negativity for protein of interest. In the case of GPB, in which the protein was not expressed in undifferentiated cells, blind sorting was conducted to derive single clones. These clones were subsequently differentiated and analysed by flow cytometry to confirm the absence of GPB surface expression in reticulocytes.

Since the founder BEL-A cell line was derived from a donor of Rh type D + C − c + E + e + (for full blood group genotype see Table 2), both RhD and RhCE required removal. To simplify this, a single guide was designed to target *RHAG*, the gene encoding the Rh-associated glycoprotein, essential for stable surface expression of Rh and in which mutations are known to result in $Rh_{null}$ (regulator type) erythrocytes (Cherif-Zahar *et al*, 1996; Huang, 1998). Individual blood group knockout BEL-A clones were expanded and a proportion

differentiated for verification of null phenotypes in enucleated reticulocytes. Figure 1 shows the complete absence of expression of proteins targeted by CRISPR gRNAs in knockout reticulocytes compared to reticulocytes derived from unedited cells, as assessed by flow cytometry. In each case, reticulocytes were also labelled with a panel of antibodies to additional major erythrocyte membrane proteins: band 3, GPA, GPC and CD47 (Fig EV1). The anticipated reduction in CD47, previously reported for $Rh_{null}$ erythrocytes (Mouro-Chanteloup *et al*, 2003), was recapitulated whilst expression of band 3, GPA and GPC was unaltered compared to untransduced control cells.

## Combinatorial CRISPR editing for generation of RBC with multiple blood group proteins ablated

Having determined guide sequences that enable the successful knockout of each of the blood groups of interest, clonal GPB null BEL-A cells were used as a starting point to generate a cell line transduced with multiple guides resulting in a multiple knockout. GPB null cells were transduced with three lentiCRISPRv2 lentiviruses simultaneously, each containing a guide targeting either *KEL* (Kell gene), *ACKR1* (Duffy gene) or *FUT1* (gene encoding fucosyltransferase 1, the enzyme required for the generation of the H antigen). Cells were immunolabelled with antibodies specific for each targeted protein, a triple null population was single cell sorted, and cells were expanded and differentiated for verification of the null phenotype in reticulocytes as described for single knockouts. Cells deficient in GPB, H, Kell and Duffy were subsequently transduced with lentiCRISPRv2 containing a guide targeting *RHAG* to produce a 5× knockout (KO) BEL-A line. Biallelic mutations in each of the genes targeted were confirmed and are listed in Table EV1.

Reticulocytes were generated by *in vitro* culture, 5× KO and control cells were induced to undergo differentiation and after 14 days reticulocytes were isolated by leukofiltration. Figure 2A shows flow cytometry histograms illustrating the absence of GPB (U and s antigen on a S− background), Kell, Duffy, H antigen, RhAG

[§]Correction added on 9 May 2018 after first online publication: the blood group nomenclature in Table 2 has been corrected.

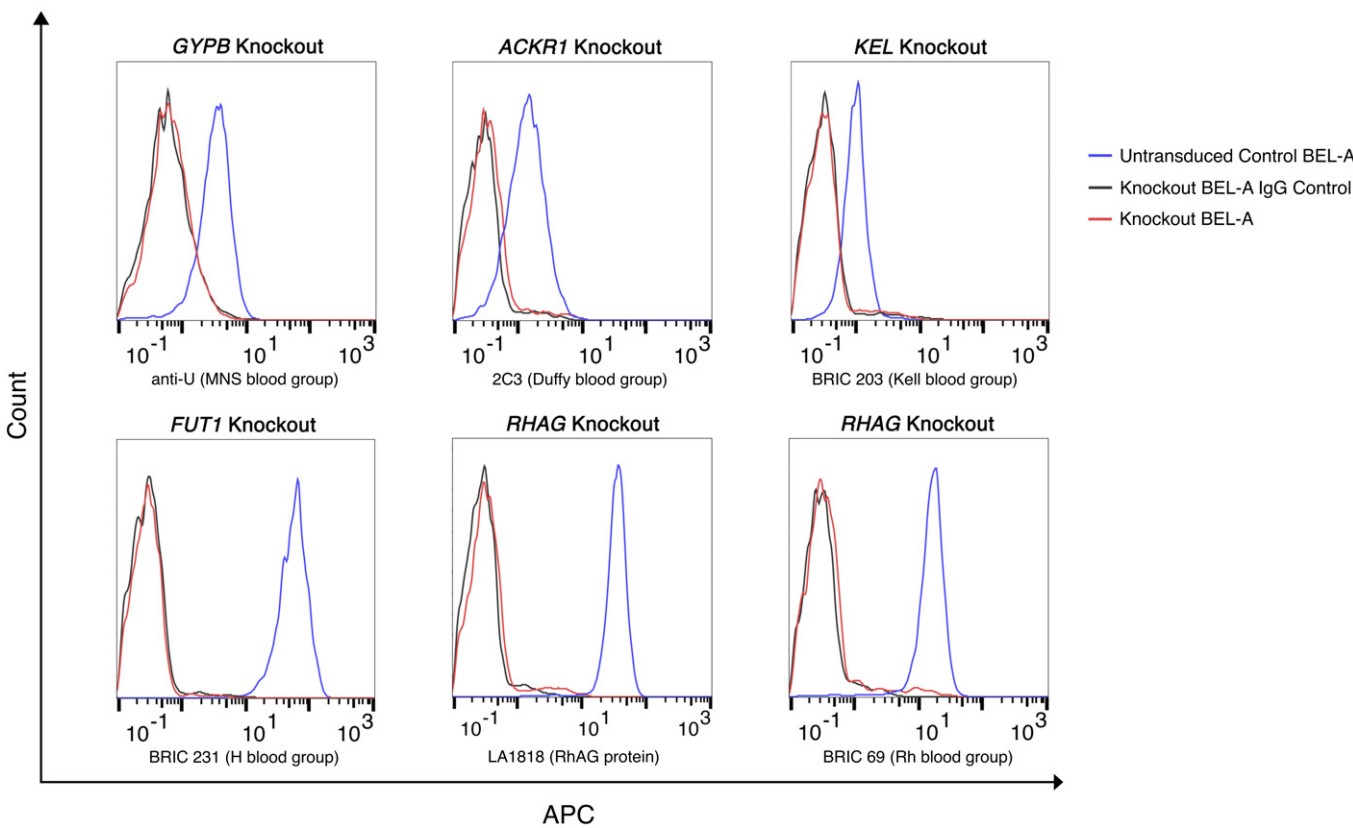

**Figure 1.    Flow cytometric confirmation of individual blood group knockouts in BEL-A-derived reticulocytes.**

BEL-A blood group knockout lines were created using lentiviral CRISPR–Cas9. Knockout lines were single cell sorted into clonal sub-lines which were differentiated for 14 days. Enucleated reticulocytes were identified based on negativity for Hoechst 33342. Expression levels of targeted blood group antigens in knockout lines overlay with IgG controls indicating complete protein knockouts. *RHAG* knockout was screened with both LA1818 (anti-RhAG) and BRIC 69 (anti-Rh) in order to confirm *RHAG* knockout and $Rh_{null}$ phenotype.

and Rh (RhCE/D). Cytospins were prepared, and 5× KO and control cells were observed to be morphologically indistinguishable (Fig 2B). Null phenotypes were confirmed by indirect antiglobulin tests (IATs) using human sera containing alloantibodies to antigens in each blood group as indicated (Fig 2C). An extended figure depicting additional RBC controls can be viewed in Fig EV2. Label-ling with a panel of antibodies to other proteins identified the expected reduction in CD47 (Mouro-Chanteloup *et al*, 2003) but no alteration in levels of band 3, GPA or GPC (Fig EV3) confirming the absence of gross membrane disruption. In order to determine whether removal of the five blood group proteins altered the physi-cal characteristics of reticulocytes, an Automated Rheoscope and Cell Analyser (ARCA) was used to assess deformability. The deformability index of 5× KO reticulocytes was found to be only mildly reduced compared to untransduced control reticulocytes (Fig 2D).

For complete protein level assessment of 5× KO cells, quantitative proteomic analysis was performed using tandem mass tag (TMT) labelled reticulocyte samples. Anticipated reductions in the Rh complex proteins CD47 and ICAM-4 (39 and 93% reductions, respec-tively) and the Kell interacting protein XK (37% reduction) were con-firmed. However, only 2% of membrane and cytoskeletal proteins showed at least a twofold reduction in abundance, demonstrating minimal structural disruption resulting from blood group protein

removal. The similarity of the two cell types is illustrated in Fig 2E which displays the $\log_2$ fold change of membrane and cytoskeletal protein abundance in 5× KO compared to control BEL-A-derived reticulocytes and Table 3 which lists the fold change of a selection of other blood group, membrane and cytoskeletal proteins important for RBC structure and function. Surprisingly, whilst flow cytometry and serology confirmed the absence of Rh at the surface, trace abun-dance of peptides was detected for Rh. Since $Rh_{null}$ phenotype of the regulator subtype was generated by targeting of the *RHAG* gene, we assume that this signal originates from intracellular fragments of Rh protein within a cellular protein degradation compartment. The complete comparative proteome of the 5× KO and control reticulo-cytes can be viewed in Fig EV4 and the Dataset EV1.

Off-target mutations arising from CRISPR–Cas9-mediated gene editing are a widely reported phenomenon (Zhang *et al*, 2015). For the generation of *in vitro* derived RBCs, the importance of these off-target mutations is mitigated by the stringent selection imparted by the terminal differentiation and enucleation processes for the genera-tion of a functional reticulocyte. Nevertheless, in order to assess the frequency and identity of off-target mutations within the clonal 5× KO BEL-A line, whole genome sequencing was performed on untransduced control and 5× KO lines (for access to raw data see, Data Availability). Mutations in 5× KO cells were identified, and the corresponding wild-type sequences were extended (100 bp upstream

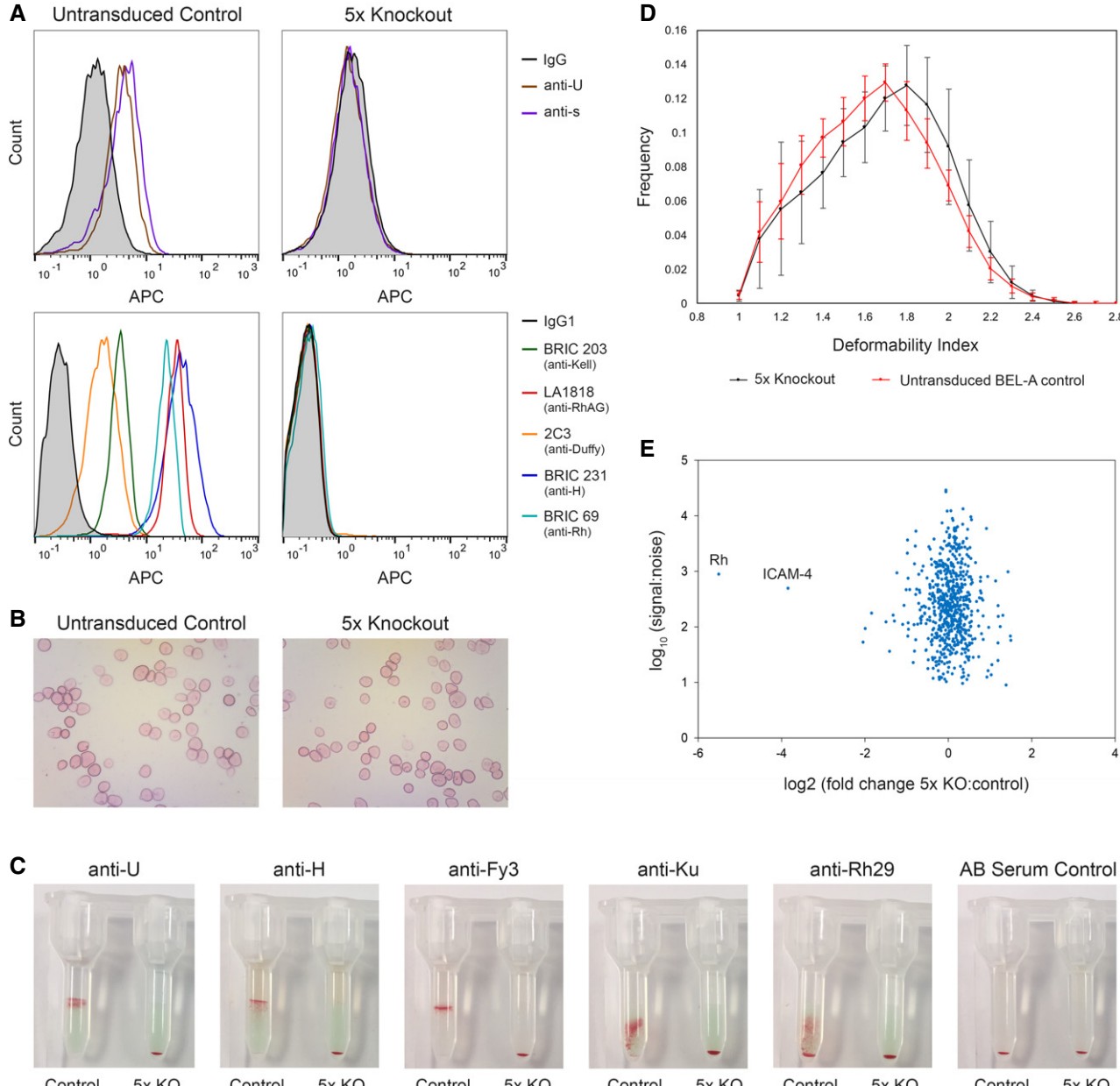

**Figure 2. Characterisation of 5× KO BEL-A reticulocyte phenotype.**

The 5× KO BEL-A cell line was created using lentiviral CRISPR–Cas9 targeted to five blood group genes: *KEL*, *RHAG*, *ACKR1*, *FUT1* and *GYPB*. Knockout cells were sorted into clonal sub-lines which were differentiated for 14 days to generate reticulocytes for analysis.

A Hoechst-negative untransduced BEL-A-derived reticulocytes are positively labelled by antibodies to indicated blood groups/antigens. 5× KO reticulocytes labelled with antibodies to targeted blood group proteins/antigens are completely deficient in expression.

B Representative cytospin images illustrate similar morphology of leukofiltered reticulocytes derived from untransduced and 5× KO BEL-A cell lines.

C Indirect antiglobulin test using column agglutination of BEL-A reticulocytes. Absence of GPB, H antigen, Duffy, Kell and Rh in 5× KO reticulocytes is supported by IAT tests with anti-U, anti-H, anti-Fy3, anti-Ku and anti-Rh29 antibodies, respectively. In contrast to the untransduced control cells, 5× KO cells did not agglutinate upon exposure to any of the tested antibodies and cell pellets were observed at the bottom of the microtubules in all tests.[#]

D Deformability index of untransduced control and 5× KO BEL-A cell line-derived reticulocytes determined using an Automated Rheoscope Cell Analyser. Untransduced BEL-A control *n* = 11, 5× KO *n* = 5. Error bars indicate standard deviation.

E Scatter plot depicting relative protein abundance of membrane and cytoskeletal proteins in reticulocytes derived from 5× KO compared to untransduced BEL-A cells as identified by TMT labelling and mass spectrometry. Data were categorised to identify membrane and cytoskeletal proteins using Proteome Discoverer 2.1. Log$_2$ fold ratios are based on the mean of two technical replicates. Data were filtered using a FDR of 1% with exclusion of proteins for which only a single peptide was detected.

[#]Correction added on 9 May 2018 after first online publication: the label of the serum control in panel C has been corrected.

**Table 3.   Quantitative proteomic analysis of functionally relevant red blood cell proteins.**

| Accession | Description | Gene ID | Coverage | No. of Peptides | No. of PSMs | No. of unique Peptides | Average abundance ratio 5×KO:UT control |
|---|---|---|---|---|---|---|---|
| P02549 | Spectrin alpha chain, erythrocytic 1 | SPTA1 | 61 | 136 | 599 | 136 | 0.96 |
| P11277 | Spectrin beta chain, erythrocytic | SPTB | 65 | 128 | 532 | 4 | 0.96 |
| P16157 | Ankyrin-1 | ANK1 | 51 | 77 | 347 | 77 | 1.02 |
| P35611 | Alpha-adducin | ADD1 | 39 | 21 | 55 | 20 | 1.01 |
| P35612 | Beta-adducin | ADD2 | 50 | 31 | 73 | 19 | 1.28 |
| P11171 | Protein 4.1 | EPB41 | 45 | 35 | 178 | 16 | 1.02 |
| P16452 | Erythrocyte membrane protein band 4.2 | EPB42 | 44 | 27 | 89 | 27 | 1 |
| Q00013 | 55 kDa erythrocyte membrane protein | MPP1 | 55 | 21 | 52 | 21 | 1.2 |
| Q08495 | Dematin | DMTN | 51 | 20 | 57 | 20 | 1 |
| P28289 | Tropomodulin-1 | TMOD1 | 47 | 15 | 32 | 15 | 1.03 |
| B4DW52 | Beta actin | ACTB | 55 | 14 | 152 | 1 | 0.86 |
| E2RVJ0 | Anion exchanger 1 (band 3) | SLC4A1 | 39 | 29 | 267 | 29 | 1.04 |
| A0A0C4DFT7 | Glycophorin-A | GYPA | 37 | 5 | 19 | 5 | 1 |
| P04921 | Glycophorin-C | GYPC | 27 | 2 | 31 | 2 | 0.99 |
| P27105 | Erythrocyte band 7 integral membrane protein | STOM | 52 | 14 | 102 | 14 | 1.35 |
| E7EMK3 | Flotillin-2 | FLOT2 | 35 | 15 | 32 | 15 | 0.98 |
| Q59GX2 | Solute carrier family 2 (Facilitated glucose transporter) Glut1 | SLC2A1 | 13 | 9 | 86 | 9 | 1.14 |
| C8C504 | Beta-globin | HBB | 94 | 21 | 2,705 | 1 | 0.93 |
| P00915 | Carbonic anhydrase 1 | CA1 | 72 | 16 | 370 | 2 | 1.27 |
| H7BY55 | Complement decay-accelerating factor | CD55 | 9 | 6 | 12 | 6 | 1.55 |
| E9PR17 | CD59 glycoprotein | CD59 | 23 | 3 | 8 | 3 | 1.44 |
| P35613 | Basigin | BSG | 36 | 10 | 22 | 10 | 1.23 |
| A0A068W6W9 | Urea transporter | SLC14A1 | 5 | 2 | 2 | 2 | 0.94 |
| A0A068W6H0 | Basal cell adhesion molecule (Lutheran blood group) | BCAM | 8 | 4 | 4 | 4 | 0.69 |
| P51811 | Membrane transport protein XK | XK | 11 | 5 | 9 | 5 | 0.63 |
| Q08722 | Leukocyte surface antigen CD47 | CD47 | 6 | 2 | 4 | 2 | 0.61 |
| P16070 | CD44 antigen | CD44 | 3 | 3 | 5 | 3 | 0.51 |
| Q14773 | Intercellular adhesion molecule 4 | ICAM4 | 14 | 3 | 4 | 3 | 0.07 |
| B2LR44 | Blood group Rh(D) polypeptide | RHD | 4 | 2 | 6 | 2 | 0.02 |

Table listing relative abundance of selected blood group, membrane and cytoskeletal proteins relevant to red blood cell immunogenicity and structural integrity based on data set summarised in Fig EV4.

and 100 bp downstream) and screened for similarity to guide sequences. Aside from the on-target mutations listed in Table EV1, no somatic single nucleotide polymorphisms (SNPs) or indels unique to the 5× KO cell line were identified that could be attributed to the CRISPR–Cas9 editing process.

## Discussion

Blood for transfusion purposes can be difficult to source for patients with rare blood group phenotypes and for patients who require repeated blood transfusions, as extended donor–recipient matching is required at the level of minor blood group antigens. Failure to identify or locate suitable donors results in an unmet clinical need that can have serious implications for the care of patients. Incompatible blood transfusions may result in delayed haemolysis with renal failure and poor haemoglobin increment (Gardner et al, 2015).

One of the major focuses of the RBC research community is to generate in vitro derived RBCs to supplement or complement the donation system where clinical needs are not currently met. The first recipients of an in vitro derived transfusion product are anticipated to be those for whom blood matching is difficult or impossible

to achieve within the donor population. An indefinite supply of RBCs could in principle be generated by immortalising $CD34^+$ haematopoietic stem and progenitor cells from individuals with rare blood types, should appropriate donors be identified. However, a more flexible approach is to use gene editing to generate customised cells combining multiple null phenotypes to broaden the transfusion compatibility of *in vitro* derived RBCs. Here, we have demonstrated the feasibility of this approach by using CRISPR–Cas9 editing of a recently published immortalised erythroblast cell line to generate customised RBC antigen phenotypes through individual or combinatorial knockout of blood group genes.

We have generated a selection of individual blood group null cells which may be used for diagnostic applications. This will likely represent the first practical use of such cells. As an expanded library, these and other knockouts to be generated in future have the potential to provide a sustainable alternative to supplement existing panels of reagent RBCs for identification of serum alloantibodies. More advanced or bespoke applications for the enhancement of diagnostic sensitivity could also be envisaged, for example removal of antigens such as those within the Knops (CR1) and John Milton Hagen (JMH) systems that mask clinically significant antigens (Poole & Daniels, 2007).

In addition to their diagnostics applications, erythroblast cell lines capable of generating RBCs deficient in individual blood groups such as $Rh_{null}$ have potential clinical use for recipients with specific rare phenotypes. The ultimate purpose of this study however was to demonstrate the power of CRISPR–Cas9 to make several clinically relevant knockouts within a single cell line that together can be tolerated to generate reticulocytes theoretically capable of servicing the widest unmet transfusion requirements. The concept of a truly universal donor RBC phenotype is a controversial one. A cell completely deficient in all known blood antigens is unlikely to be viable given the key structural roles played by several blood group proteins. Therefore, careful consideration of target antigens for removal is required. However, priorities for the removal of blood groups to enhance donor compatibility vary with prevalence of antigens within different ethnic populations. The RBC we have generated, null for five blood groups, would be theoretically capable of servicing the unmet clinical need of 48 of the 56 challenging transfusion cases identified by the NHSBT in England over a 15-month period. Further alterations are possible albeit with diminishing gains in compatibility. For example, the additional removal of Lutheran and Kidd would have each serviced an additional three patients.

Whilst individual absence of four of the five blood group proteins edited in this study occurs naturally without pathological effects (Anstee, 2011), the $Rh_{null}$ phenotype is associated with mild compensated anaemia (Sturgeon, 1970). Therefore, for transfusion purposes, the enhanced compatibility provided by complete Rh protein ablation may not outweigh the increased osmotic fragility and reduced circulatory half-life reported for naturally occurring $Rh_{null}$ erythrocytes (Ballas *et al*, 1984; Cartron, 1999). Interestingly, and contrary to expectations, we observed that 5× KO reticulocytes exhibited only slightly reduced cellular deformability compared to unedited BEL-A-derived reticulocytes. In the light of observations made in previous studies, it is possible that *in vivo* circulatory maturation, incompletely recapitulated by *in vitro* culture at present, is required to reveal such phenotype. Future efforts towards the

development of more practical *in vivo* systems for monitoring long-term reticulocyte survival and maturation will be informative in this regard. Rather than complete ablation of RhAG or RhCE/D, maximisation of compatibility in future development of this work could be achieved through careful selection of $CD34^+$ cells with a specific Rh antigen profile prior to immortalisation of a new founder BEL-A line potentially in combination with homology directed repair-based editing of specific antigenic residues.

The translation of genetically engineered BEL-A lines into clinical products for transfusion still faces many challenges. Improved scalability, reductions in cost of culture, and production of a GMP compatible line are needed. Alternative transient approaches to CRISPR–Cas9 expression, and more specific targeting of antigens to lessen disruption to the membrane, each offer advantages that may enhance a future product. Nevertheless, the application of gene editing to the manufacture of RBCs from immortalised erythroid progenitor cell lines ushers closer an era in which the transfusion needs of patients with rare blood group phenotypes will be met using erythroblast cell lines. We present here a major step towards the generation of customisable or multi-compatible RBCs that alongside developments in efficiency of *in vitro* erythroid culture could ultimately facilitate transfusion of patients with unmet clinical needs.

## Materials and Methods

### BEL-A cell culture

BEL-A cells were cultured as previously described (Trakarnsanga *et al*, 2017). In brief, cells were maintained in expansion medium [StemSpan SFEM (Stem Cell Technologies) supplemented with 50 ng/ml SCF, 3 U/ml EPO, 1 μM dexamethasone and 1 μg/ml doxycycline] at $1–3 × 10^5$ cells/ml. Complete medium changes were performed every 48 h. Differentiation was induced as previously described (Trakarnsanga *et al*, 2017) with modifications as follows: cells were seeded at $2 × 10^5$/ml in differentiation medium supplemented with 1 ng/ml IL-3, 10 ng/ml SCF and 1 μg/ml doxycycline. After 2 days, cells were reseeded at $3.5 × 10^5$/ml in fresh medium. On differentiation day 4, cells were reseeded at $5 × 10^5$/ml in fresh medium without doxycycline. On differentiation day 6, a complete media change was performed and cells were reseeded at $1 × 10^6$/ ml. On day 8, cells were transferred to differentiation medium (containing no SCF, IL-3 or doxycycline) and maintained at $1 × 10^6$/ ml with complete medium changes every 2 days until day 14.

### Lentiviral transduction of single guide CRISPR–Cas9 vectors

Pre-validated CRISPR guides to blood group genes in the lentiviral vector lentiCRISPRv2 were ordered from GenScript (Sanjana *et al*, 2014). gRNA sequences used are listed in Table EV2. Lentivirus was prepared according to previously published protocols (Satchwell *et al*, 2015). For transduction of BEL-A cells, virus was added to $2 × 10^5$ cells in 2 ml medium in the presence of 8 μg/ml polybrene for 24 h. Cells were washed, and resuspended in fresh medium, and after 24 h, transduced cells were selected using 1 μg/ml puromycin for 48 h. Following further expansion, cells were sorted based on antigen null phenotype to derive clones.

### Flow cytometry and FACS

For flow cytometry on undifferentiated BEL-As, $1 \times 10^5$ cells resuspended in PBSAG (PBS + 1 mg/ml BSA, 2 mg/ml glucose) + 1% BSA were labelled with primary antibody for 30 min at 4°C. Cells were washed in PBSAG, incubated for 30 min at 4°C with appropriate APC-conjugated secondary antibody, and washed and data acquired on a MacsQuant VYB Analyser using a plate reader. For FACS sorting of immunolabelled cells, a BDInflux Cell Sorter was used to isolate single clones from the antigen null, propidium iodide-negative population into 96-well plates. For differentiated BEL-As, cells were stained with 5 μg/ml Hoechst 33342 then fixed if required in 1% paraformaldehyde, 0.0075% glutaraldehyde to reduce antibody binding-induced agglutination before labelling with antibodies as described. Reticulocytes were identified by gating upon Hoechst-negative population.

### Antibodies

Mouse monoclonal antibodies were used as follows: (1:2 dilutions of unpurified supernatant), BRIC231 (anti-H) LA1818 (RhAG), BRIC69 (RhD/CE), BRIC203 (Kell), BRIC4 (GPC), BRIC32 (CD47), BRIC222 (CD44), BRIC256 (GPA) (all IBGRL), 2C3 (FY) (INSERM, Paris). Human antibodies were used for detection of s antigen (Sanquin Blood Supply, 1:50) and U antigen (IBGRL, 1:5). Secondary antibodies (1:50) were APC-conjugated monoclonal anti-mouse IgG1 or polyclonal anti-IgG (Biolegend) or Alexa647-anti-human (Jackson Laboratories).

### Serological detection of blood group antigens

Human anti-H, anti-U, anti-Rh29, anti-Ku and mouse monoclonal anti-Fy3 (MIMA29, New York Blood Centre; all antibodies were derived from the IBGRL reference collection) were used for serological detection of blood group antigens. Cell suspensions were prepared from $0.5 \times 10^6$ or $1.0 \times 10^6$ BEL-A-derived reticulocytes and pelleted at 500 *g* for 5 min; supernatant was removed, and reticulocytes were resuspended in 20 μl of ID-Diluent 2 (Bio-Rad Laboratories, Switzerland). All antibodies were tested by IATs using column agglutination technology. Each human antibody was tested in Bio-Rad LISS/Coombs ID-Cards, and MIMA29 was tested in Bio-Rad ID-PNH Test Cards (both Bio-Rad Laboratories, Switzerland). 20 μl of prepared cell suspension was added to each column followed by 10 μl of the relevant antibody or antisera. Cards were incubated for 15 min at 37°C and centrifuged as per manufacturer instructions.

### Verification of CRISPR on- and off-target effects by whole genome sequencing

For verification of CRISPR edits, genomic DNA was isolated from specific clones using a DNeasy Blood and Tissue Kit (Qiagen). DNA quality validation, whole genome sequencing and bioinformatic analysis were performed by Novogene. Genomic DNA was isolated from unedited control or 5× KO cells using a DNeasy Blood and Tissue Kit (Qiagen), checked for purity using a NanoPhotometer spectrophotometer (Implen), and DNA concentration was measured using Qubit DNA Assay Kit in a Qubit 2.0 Fluorometer (Life Technologies). A total amount of 1.0 μg genomic DNA per sample was used as input material for the DNA sample preparations. RNA was removed using RNase at 37°C for 25 min. Sequencing libraries were generated using NEBNext® DNA Library Prep Kit following manufacturer's recommendations. The genomic DNA was randomly fragmented to a size of 350 bp by shearing, and then, DNA fragments were end polished, A-tailed, and ligated with the NEBNext adapter for Illumina sequencing, and further PCR enriched by P5 and indexed P7 oligos. The PCR products were purified (AMPure XP system), and resulted libraries were analysed for size distribution by Agilent 2100 Bioanalyzer and quantified using real-time PCR.

The clustering of the index-coded samples was performed on a cBot Cluster Generation System using Hiseq X HD PE Cluster Kit (Illumina) according to the manufacturer's instructions. After cluster generation, the library preparations were sequenced on an Illumina Hiseq X Ten platform and paired-end reads were generated. The original raw data were transformed to sequenced reads by base calling and recorded in FASTQ file, which contains sequence information (reads) and corresponding sequencing quality information. Burrows–Wheeler Aligner (BWA) was utilised to map the paired-end clean reads to the human reference genome (b37, ftp://gsapubftp-anonymous@ftp.broadinstitute.org/bundle/b37/human_g1k_v37_decoy.fasta.gz). SAMtools and Picard were used to perform BAM sorting and duplicate marking to generate final BAM file for computation of the sequence coverage and depth. A total of 90.6 Gb clean reads with average sequencing depth of 30.2× and a whole genome coverage of 99% (bases with depth > 10 × 98.1%) were obtained for the unedited cell sample and for the 5× KO sample, 194.2 Gb with a depth of 65.05× and coverage of 99.1% (98.9% with depth > 10×).

We focused on the potential off-target effect of Cas9 and five gRNAs expression within the whole genome by comparing the genomic differences between the unedited control and 5× KO samples. For both samples, SNPs were detected by muTect (Cibulskis *et al*, 2013), the indels by Strelka (Saunders *et al*, 2012). All somatic SNPs and indels (up to 50 nt) unique to the 5× KO cells were identified by comparison with the parent line. These mutation sequences were extended to include 100 bp upstream and 100 bp downstream of the mutation sites, the corresponding wild-type sequences were extracted, and a BLAST search was performed using each gRNA + PAM sequence to identify off-target mutations. Results were filtered based on the editing characteristics of the CRISPR–Cas9 to identify mutations attributable to the editing process according to criteria in which (i) the cleavage site was proximal to 3 bp upstream of the PAM and (ii) less than 6 bp of mismatch exists between gRNA and potential gRNA target host genome plus PAM ("NRG"), or a continuous match of the final 10-bp seed sequence immediately upstream of the PAM NRG occurs.

### TMT labelling, mass spectrometry and data analysis

Total cell lysates from $2 \times 10^6$ reticulocytes per sample were digested with trypsin (2.5 μg trypsin per 100 μg protein; 37°C, overnight) and labelled with Tandem Mass Tag (TMT) six plex reagents according to the manufacturer's protocol (Thermo Fisher Scientific, Loughborough, LE11 5RG, UK), and the labelled samples were pooled. An aliquot of the pooled sample was evaporated to dryness, resuspended in 5% formic acid and then desalted using SepPak cartridges according to the manufacturer's instructions (Waters, Milford, MA, USA). Eluate from the SepPak cartridge was again evaporated to

dryness and resuspended in buffer A (20 mM ammonium hydroxide, pH 10) prior to fractionation by high pH reversed-phase chromatography using an Ultimate 3000 liquid chromatography system (Thermo Fisher Scientific). In brief, the sample was loaded onto an XBridge BEH C18 Column (130 Å, 3.5 μm, 2.1 × 150 mm, Waters, UK) in buffer A and peptides eluted with an increasing gradient of buffer B (20 mM ammonium hydroxide in acetonitrile, pH 10) from 0 to 95% over 60 min. The resulting fractions were evaporated to dryness and resuspended in 1% formic acid prior to analysis by nano-LC MSMS using an Orbitrap Fusion Tribrid mass spectrometer (Thermo Scientific).

High pH RP fractions were further fractionated using an Ultimate 3000 nano-LC system in line with an Orbitrap Fusion Tribrid mass spectrometer (Thermo Scientific). In brief, peptides in 1% (vol/vol) formic acid were injected onto an Acclaim PepMap C18 nano-trap column (Thermo Scientific). After washing with 0.5% (vol/vol) acetonitrile 0.1% (vol/vol) formic acid, peptides were resolved on a 250 mm × 75 μm Acclaim PepMap C18 reverse-phase analytical column (Thermo Scientific) over a 150-min organic gradient, using seven gradient segments (1–6% solvent B over 1 min, 6–15% B over 58 min, 15–32% B over 58 min, 32–40% B over 5 min, 40–90% B over 1 min, held at 90% B for 6 min and then reduced to 1% B over 1 min) with a flow rate of 300 nl/min. Solvent A was 0.1% formic acid, and Solvent B was aqueous 80% acetonitrile in 0.1% formic acid. Peptides were ionised by nano-electrospray ionisation at 2.0 kV using a stainless-steel emitter with an internal diameter of 30 μm (Thermo Scientific) and a capillary temperature of 275°C.

All spectra were acquired using an Orbitrap Fusion Tribrid mass spectrometer controlled by Xcalibur 2.0 software (Thermo Scientific) and operated in data-dependent acquisition mode using an SPS-MS3 workflow. FTMS1 spectra were collected at a resolution of 120,000, with an automatic gain control (AGC) target of 400,000 and a max injection time of 100 ms. Precursors were filtered with an intensity range from 5,000 to 1E20, according to charge state (to include charge states 2–6) and with monoisotopic precursor selection. Previously interrogated precursors were excluded using a dynamic window (60s ± 10 ppm). The MS2 precursors were isolated with a quadrupole mass filter set to a width of 1.2 m/z. ITMS2 spectra were collected with an AGC target of 10,000, max injection time of 70 ms and CID collision energy of 35%.

For FTMS3 analysis, the Orbitrap was operated at 30,000 resolution with an AGC target of 50,000 and a max injection time of 105 ms. Precursors were fragmented by high energy collision dissociation (HCD) at a normalised collision energy of 55% to ensure maximal TMT reporter ion yield. Synchronous precursor selection (SPS) was enabled to include up to five MS2 fragment ions in the FTMS3 scan.

The raw data files were processed and quantified using Proteome Discoverer software v2.1 (Thermo Scientific) and searched against the UniProt Human database (134,169 entries) using the SEQUEST algorithm. Peptide precursor mass tolerance was set at 10 ppm, and MS/MS tolerance was set at 0.6 Da. Search criteria included the oxidation of methionine (+15.9949) as a variable modification and carbamidomethylation of cysteine (+57.0214) and the addition of the TMT mass tag (+229.163) to peptide N-termini and lysine as fixed modifications. Searches were performed with full tryptic digestion and a maximum of one missed cleavage was allowed. The reverse database search option was enabled, and all peptide data were filtered to satisfy false discovery rate (FDR) of 1%.

### The paper explained

#### Problem

The provision of blood for patients who require repeated blood transfusions, as well as for individuals with rare blood types, presents an enormous challenge to transfusion services worldwide. Patients with RBC disorders such as thalassaemia or sickle-cell disease require regular transfusions. With repeated transfusion, alloimmunisation often occurs due to incompatibility at the level of minor blood group antigens. Individuals with rare blood group phenotypes, such as those with the Bombay phenotype and Rh$_{null}$ individuals, require difficult-to-source blood from individuals with the same rare phenotype otherwise risk haemolytic transfusion reactions.

#### Results

We have generated a selection of erythroblast cell lines with key blood group proteins knocked out which can be differentiated to produce reticulocytes. Furthermore, we have combined knockouts to produce a single cell line which can generate reticulocytes with absence of multiple blood group proteins.

#### Impact

The generation of *in vitro* derived RBCs to supplement or complement the donation system is a major focus of the RBC research community. This is the first study to generate deformable reticulocytes from an immortalised adult erythroblast cell line which has been genetically edited using CRISPR–Cas9 to improve transfusion compatibilities. The first recipients of an *in vitro* derived transfusion product are anticipated to be those for whom alloimmunisation or rare blood group phenotypes make blood matching difficult or impossible to achieve within the donor population. We have demonstrated CRISPR-mediated gene editing as a flexible approach to the successful production of *in vitro* derived RBCs with customised or unique blood group phenotypes that could ultimately facilitate transfusion of patients with unmet clinical needs.

### Reticulocyte deformability measurements using ARCA

$1 \times 10^6$ BEL-A-derived reticulocytes were resuspended in 200 μl polyvinylpyrrolidone solution (PVP viscosity 28.1; Mechatronics Instruments, The Netherlands). Samples were assayed in an ARCA (Dobbe *et al*, 2002) consisting of a plate–plate optical shearing stage (model CSS450) mounted on a Linkam imaging station assembly and temperature controlled using Linksys32 software (Linkam Scientific Instruments, Surrey, UK). The microscope was equipped with an LMPlanFL 50× with a 10.6 mm working distance objective (Olympus, Essex, UK) illuminated by a X-1500 stroboscope (PerkinElmer, The Netherlands) through a band-pass interference filter (CWL 420 nm, FWHM 10 nm; Edmund Optics, Poppleton, UK). Images were acquired using a uEye camera (UI-2140SE-M-GL; IDS GmbH, Obersulm, Germany). At least 1,000 cell images per sample were acquired and analysed using bespoke ARCA software.

### Data availability

The mass spectrometry proteomics data from this publication have been deposited to the ProteomeXchange Consortium via the PRIDE (Vizcaino *et al*, 2016) partner repository with the data set identifier PXD009291. The whole genome sequencing data are available at https://doi.org/10.5523/bris.3owu3wwvhghy5278tzd 6b8wutz.

Expanded View for this article is available online.

## Acknowledgements

The authors wish to acknowledge the assistance of Dr. Andrew Herman and Lorena Sueiro Ballesteros of University of Bristol flow cytometry facility for cell sorting and Dr. Rosey Mushens for preparation and provision of monoclonal antibodies. We thank Dr Wassim El Nemer (Inserm, Paris) for the gift of the 2C3 FY antibody. This research was funded by grants from NHS Blood and Transplant R&D committee (NHSBT WT15-04 and WT15-05; DJA and AMT), a National Institute for Health (NHIR) grant to support a NIHR Research Blood and Transplant Unit (NIHR BTRU) in Red Blood Cell Products at the University of Bristol in Partnership with NHSBT (NIHR-BTRU-2015-10032; TJS, JF, DJA, AMT). JH was funded by a EPSRC/BBSRC SynBio Centre CDT PhD with Defence Science and Technology Laboratory as an industrial partner and DD by a PhD from NIHR BTRU. MM was funded by BrisSynBio via a BBSRC/EPSRC Synthetic Biology Research Centre Grant (BB/L01386X/1; to AMT). The views expressed are those of the author(s) and not necessarily those of the NHS, the NIHR or the Department of Health.

## Author contributions

TJS and JH contributed equally to this work. Experiments were conceived and designed by TJS, JH and AMT with contribution from DJA and JF. TJS and JH carried out the majority of experiments, performed the analysis and prepared the figures. MM and DED conducted experiments. NMT conducted serological testing. MCW and KH performed proteomics analysis. JGGD and GJS provided essential equipment and analysis software. KT provided original BEL-A cell line and culture methodology. FR conducted the patient survey and provided transfusion knowledge. TJS, JH and AMT wrote the manuscript. All authors read and edited the manuscript.

## Conflict of interest

The authors declare that they have no conflict of interest.

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
