## [Review Process File · EMBO Molecular Medicine]

Enhancement of red blood cell transfusion compatibility using CRISPR-mediated erythroblast gene editing

J Hawsworth, TJ Satchwell, M Meinders, DE Daniels, F Regan, NM Thornton, MC Wilson, J.G.G Dobbe, GJ Streekstra, K Trakarsanga, KJ Heesom, DJ Anstee, J Frayne, AM Toyé

Review timeline:

Submission date:	4 September 2017
Editorial Decision:	6 September 2017
Appeal received:	6 September 2017
Editorial Decision:	9 October 2017
Additional correspondence	13 December 2017
Additional correspondence	14 December 2017
Revision received:	21 February 2018
Editorial Decision:	8 March 2018
Revision received:	22 March 2018
Accepted:	23 March 2018

Editor: Céline Carret

Transaction Report:

1st Editorial Decision

6 September 2017

Thank you for the submission of your manuscript "Enhancement of red blood cell transfusion compatibility using CRISPR-mediated erythroblast gene editing". I have now had the opportunity to carefully read your paper and the related literature and I have also discussed it with my colleagues. I am afraid that we concluded that the manuscript is not well suited for publication in EMBO Molecular Medicine and have therefore decided not to proceed with peer review.

We appreciate that your data report a proof-of-principle of using the CRISPR-Cas9 system to genetically modify an immortalised red blood cell line you recently published in order to delete specific antigens to avoid alloimmunisation issues upon transfusion. While the technology is clearly of interest for the immediate field, I am afraid that the concept of modifying RBC lines genetically is not novel, albeit not with CRISPR-Cas9. In addition, the current study does not show per se that this enhanced cell line has improved transfusion capabilities. Therefore, I am afraid that we find the study not appropriate at this time to be further evaluated in EMBO Molecular Medicine.

I am sorry to have to disappoint you on this occasion; in the interest of time, I am providing you with an early decision that will allow you to submit your manuscript elsewhere without any further delays.

Please rest assured that this is not a judgment of the quality or interest of your work but a decision based on appropriateness for EMBO Molecular Medicine.

Appeal

6 September 2017

Thank you for your quick reply. I have to say I am extremely surprised and disappointed that you do not think it's an appropriate paper given previous publications in EMBO Molecular Medicine - in particular I flag the Vel blood group paper in my cover letter.

This manuscript is the first ever example that shows individual and combined multiple blood group knockouts using CRISPR in a single cell line - removing not only ABO blood group (H-antigen) to make Bombay blood type but also four other clinically problematic blood groups including Rhesus. This has never been done before and is a first for transfusion medicine. This is an engineering first with a new red cell phenotype that keeps its structure and makes a more compatible blood product. This also has important applications in other fields including Malaria invasion as it illustrates ways of efficiently removing host receptors that are blood groups.

I would have thought this would be enough for the manuscript to be sent to review, so I am really surprised this has not happened.

The BEL-A erythroblast cell line which we edited in this study was published in Nature Comms and garnered much international interest, both scientific and public (see the article tracker on the Nature website). We anticipate this paper would also generate similar interest.

The current cell line cannot be used in humans - this is proof of principle research. The first of its kind. Human tests would need a GMP compatible cell. It's important to note that the BEL-A line is the only adult erythroid line that enucleates. IPS cells do not differentiate properly to reticulocytes (immature red cells) and other erythroid lines barely enucleate and the cells produced are unstable.

I respect your decision but is there no chance you would reconsider and send for review from the scientific community, such as the eminent transfusion experts we suggest as reviewers?

2nd Editorial Decision

9 October 2017

Thank you for the submission of your manuscript to EMBO Molecular Medicine. We have now heard back from the three referees whom we asked to evaluate your manuscript.

As you will see from the reports below, the referees find the topic of your study of interest. However, they raise substantial concerns on your work. Of particular relevance for the clinical focus of the study, we would need more functional analyses and better molecular and genomic characterisations of the resulting cells (referees 1 and 3). We realise that this is meant as a proof-of-principle article but we would need some indication that all these genetic manipulations do not have long-term side effect (as mentioned by referees 2 and 3).

We would welcome the submission of a revised version within three months for further consideration and would like to encourage you to address all the criticisms raised as suggested to improve conclusiveness and clarity. Please note that EMBO Molecular Medicine strongly supports a single round of revision and that, as acceptance or rejection of the manuscript will depend on another round of review, your responses should be as complete as possible. I note that addressing the reviewers concerns in full will be necessary for further considering the manuscript in our journal and this appears to require a lot of additional work and experimentation. I realise that you may not want to provide such a revision, and rather seek rapid publication elsewhere at this stage.

***** Reviewer's comments *****

Referee #1 (Comments on Novelty/Model System for Author):

As noted in comments below, similar results have been published in a more well characterized erythroid cell line previously (Kim et al., Nature Communications, 2015). Moreover, the characterization discussed in this manuscript in terms of molecular changes and the phenotype of mature RBCs is fairly limited.

Referee #1 (Remarks for Author):

In this manuscript from Hawksworth and colleagues, an important problem in transfusion medicine is addressed. The authors use a recently described erythroid cell line, the BEL-A cell line, to perform genome editing to alter expression of multiple blood groups in an attempt to create potential universal donor cells.

While the authors are studying an important area, this manuscript lacks both novelty and rigor. A prior study that is not cited in this article, has performed similar work in a comparable erythroid cell line (Kim et al., Nature Communications, 2015). Indeed, a number of groups have created mature red cells from primary HSCs and pluripotent stem cells. Other than the single prior publication from this same group, the BEL-A cell line has had no characterization or analysis from other groups, in contrast to the well-studied HUDEP cell lines. This publication lacks detailed characterization of the cells and it is unclear from the data shown whether these cells may ever be a useful transfusion alternative. A reasonable starting point would be to perform whole genome sequencing on the modified cells to examine global changes in the genome and to ensure that all changes are on target. Additionally, performing such genome editing experiments in primary HSCs would be valuable to demonstrate that this can be done in a primary cell setting that may more effectively be translated to the clinical setting.

Referee #2 (Comments on Novelty/Model System for Author):

The use of the immortalized BEL-A cell line for downstream modification of erythrocyte membrane proteins is very exciting. It opens up the opportunity to create tailor-made erythrocytes for transfusion to immunized patients for whom finding compatible blood is a major challenge. Whether this approach of multiple knockout will succeed in producing viable, functional erythrocytes remains to be seen however the potential cannot be ignored.

Referee #2 (Remarks for Author):

General Comments

This is a concise and exciting proof-of-principle paper to demonstrate the feasibility of using CRISPR-mediated gene editing to knock-out multiple integral erythrocyte membrane proteins without apparent gross structural damage. The premise, i.e. creation of a "universal" donor erythrocyte, is a goal for many research groups in the field, and these investigators demonstrate that it can be done, at least technically. There is so much more to be done to determine how viable these cells are in the long term, but as all work in this exciting field, the knowledge gained into the biology of is as important as achieving the end goal of ex vivo erythrocyte production for use in transfusion therapy.

The first, and perhaps major, criticism is that the title of the paper does not reflect the content. There is no evidence given to show "Enhancement of red blood cell transfusion compatibility". The authors have not demonstrated by simple haemagglutination that the 5xKO cells they have generated are indeed compatible with the plasma of at least one of the 48/56 patients for which they claim these cells would be a better transfusion alternative. I recommend either that the title is reworded or, more clinically relevant, that the cell lines are tested with representative plasma samples to give credit to the current title.

It was interesting to read on page 4 and see in Figure EV1 that expression of CD47 was considerably reduced and it will be interesting to see the long term effect of this change on cell survival.

Specific comments (note: the manuscript was not paginated and thus numbers below are assigned from the title page forward. Line numbers are given including paragraph titles, starting with Abstract)

Page 1, Line 21

There are currently 352 blood group antigens recognised by the ISBT, not "more than 400".

Page 2, lines 12-14

This sentence states that "Current efforts to cater..." then lists a reference from 1999. There are more recent references that would make this sentence more credible. Two are given here:

Nance S, Scharberg EA, Thornton N, Yahalom V, Sareneva I, Lomas-Francis C. International rare donor panels: a review. *Vox Sang*. 2016 Apr;110(3):209-18.

Nance ST. How to find, recruit and maintain rare blood donors. *Curr Opin Hematol*. 2009 Nov;16(6):503-8.

Page 7, line 3

It appears that there is an extra "+1%BSA" in this sentence describing the reagent PBSAG.

Page 11, Table 1

It was interesting to note that antigens such as C^W, K, Kp^a, and Lu^a are listed as difficult to match since these are of low prevalence. Also, the use of ISBT nomenclature is recommended for describing blood group antigens, i.e. Kp^a not Kpa, etc.

Page 13, Figure 1

The low signal given by the untransduced BEL-A cells when tested with BRIC 203 was surprising. Is there an explanation for the weak expression of the Kell glycoprotein in these cells?

Page 14, Figure 2

I recommend that the upper and lower panels in A are switched to reflect the initial knockout of GPB prior to the subsequent knockout of Duffy, RhAG, FUT and Kell.

In panel C, it would be simpler to show the Rheoscope results of the two 5xKO cells individually rather than giving standard deviation, which is not relevant in such a small sample size.

Table EV2

Please list phenotypes according to ISBT nomenclature.

Referee #3 (Comments on Novelty/Model System for Author):

The manuscript lacks any molecular analysis of the resulting cells with only one assay (deformability) to test the physical properties of the resulting cell line.

Referee #3 (Remarks for Author):

This is a very nice proof of principle paper where the authors use their recently developed BEL-A cell line to show that at least five red blood cell antigens can be knocked out without substantially changing the capacity of the cells to differentiate to enucleated RBCs and that their deformability index has hardly changed. The ability to do this is a step forward to obtain RBCs for transfusion in (rare) patients that develop an immune reaction to these RBC antigens. The method to derive these cells was a sequential KO using Crispr/Cas with different guide RNAs for the different genes. These were delivered via viral infection and stable cell lines were obtained that the authors show are convincingly negative for the targeted antigens. They subsequently show that the deformability of the RBCs obtained after differentiation of these cells is very comparable to that of wt cells. It is disappointing that the authors do not provide any molecular analysis of their KO cells. It is unknown what mutations actually occurred, whether there are bystander effects (particularly for 5 KO's). Best would have been of course to sequence the genome of the cells before and after the KO's to obtain a precise picture of the process. After all this is presented as a way to get to "universal" donor cells. A number of groups are pursuing similar or very different methods to achieve a similar goal and hence it is important to carefully describe the results even if it is only presented as a proof of principle. Thus I recommend that the authors revise the manuscript both at the level of molecular and physical property analysis

Additional correspondence (author)

13 December 2017

I wanted to let you know that we have completed the laboratory work that we needed to do. We have undertaken multiple cell cultures to produce more reticulocytes (each culture takes about 2-3 weeks).

We have collaborated with Nicole Thornton at International Blood Group Laboratory in NHSBT Filton to optimise and undertake serological examination of the reticulocytes. This has confirmed that human plasma samples which have a known antibody reaction to the individual blood groups does not react with the cells. This confirms the flow cytometry results and was conducted using gold standard methodology in terms of serology.

We are now waiting on quantitative proteomics to provide further characterization of the reticulocytes. The samples are being processed for TMT mass spectrometry in the University of Bristol Proteomics facility and will then need to be run and subjected to analysis. This will provide a comprehensive readout of erythroid proteins present in the reticulocytes of the normal and the 5x knockout. Unfortunately, the samples can not be run before Christmas due to staff holidays. The proteomics facility will be closed for 2 weeks for Christmas. They will run them as soon as they can in January.

Although this work was always proof of principle we have shown willing and undertaken whole genome sequencing of the 5xnull clone. We had to take significant time to identify a suitable company who could process, run the sequencing to an adequate depth and undertake analysis. Again due to Christmas holidays we are likely to have a delay on the results.

This makes it difficult to predict whether we will be able to deliver the results by the current 9th of January deadline especially as we have lost 2 weeks due to Christmas holidays.

Given that the 3-months revision time also includes the Christmas holidays I am writing to request an extension to the original revision time. I am not currently sure how long the whole genome sequencing will take as it is now out of our hands. I hope this work will all be completed ASAP - ideally by the end of January but I request until the end of February to be 100% sure we have enough time. If you can also extend the scooping protection that would be appreciated.

I hope you understand our reasons for requesting an extension and will be happy to grant this request-especially given that the revision period includes the Christmas holiday?

Additional correspondence (editor)

14 December 2017

Thank you for updating us on your progress. We understand that the technical issues are out of your hands. We will reset the due date to 28th February. The scooping protection will still be active.

We look forward to receiving your revised manuscript.

1st Revision - authors' response

21 February 2018

Referee #1 (Comments on Novelty/Model System for Author):

As noted in comments below, similar results have been published in a more well characterized erythroid cell line previously (Kim et al., Nature Communications, 2015). Moreover, the characterization discussed in this manuscript in terms of molecular changes and the phenotype of mature RBCs is fairly limited.

Referee #1 (Remarks for Author): In this manuscript from Hawksworth and colleagues, an important problem in transfusion medicine is addressed. The authors use a recently described erythroid cell line, the BEL-A cell line, to perform genome editing to alter expression of multiple blood groups in an attempt to create potential universal donor cells.

While the authors are studying an important area, this manuscript lacks both novelty and rigor. A prior study that is not cited in this article, has performed similar work in a comparable erythroid cell line (Kim et al., Nature Communications, 2015).

We are pleased that the reviewer believes that this is an important area of study however we must disagree with their assessment of the novelty and rigour. The concept of modifying red blood cells to improve compatibility for transfusion or for generation of ‘universal donor’ phenotypes is not a novel one and has been attempted using a variety of approaches over many years including modification of haematopoietic stem cells derived from different sources as discussed in the introduction. The fact that this concept continues to represent an active area of research for so many groups underscores its importance in transfusion medicine. Whilst not novel conceptually, the work presented here is the first study to our knowledge to combine multiple edits and blood group null phenotypes within viable cell line derived red blood cells and therefore represents a significant practical advance in the field.

The study (Kim et al, 2015) referred to by the reviewer uses TALENs to edit and ablate expression of the *RHD* gene in an erythroid cell line. The study uses the HIDEP cell line initially derived from induced pluripotent stem cells (iPSCs). The HIDEP cell line produces differentiated cells expressing fetal rather than adult globins. The HUDEP cell line also referred to by the reviewer (below), was not edited in the referenced publication. HUDEP cells are derived from umbilical cord blood and also express inappropriate globins. Furthermore, the phenotype generated by Kim and colleagues (RhD negative) is not uncommon, representing 15% of the Caucasian population and 7% and 10% of African and Asian populations respectively, and thus provides evidence only of the capacity to generate cells with established absence of phenotype and of limited practical necessity or utility in contrast to the novel complex phenotype generated in our manuscript. Thus, while there are conceptual similarities, the work within the two studies is significantly different in scope. We have nevertheless amended the manuscript to include reference to this study in the introduction.

Indeed, a number of groups have created mature red cells from primary HSCs and pluripotent stem cells. Other than the single prior publication from this same group, the BEL-A cell line has had no characterization or analysis from other groups, in contrast to the well-studied HUDEP cell lines.

The recently published BEL-A cell line uses the same immortalisation approach employed for generation of the HI- and HUDEP lines in order to produce a line with the capacity for generation of reticulocytes expressing adult globins (Trakarnsanga et al, 2017). We dispute the reviewer’s assertion that the BEL-A cells are less appropriate than HUDEP cells for this work. As a more recently developed resource (publication March 2017) the cells are yet to be widely used in other studies compared to lines that were published several years ago, however we do not believe this is an appropriate basis to ignore the advantages they present in recapitulating reticulocytes of adult phenotype. What is more, whilst the HIDEP and HUDEP cells have been used in a variety of publications, the majority of these studies focus on aspects of erythroblast expansion and the early stages of differentiation with little to no characterisation of reticulocytes. In contrast, a detailed proteomic characterisation of BEL-A derived reticulocytes was published as part of Trakarnsanga et al earlier this year.

This publication lacks detailed characterization of the cells and it is unclear from the data shown whether these cells may ever be a useful transfusion alternative. A reasonable starting point would be to perform whole genome sequencing on the modified cells to examine global changes in the genome and to ensure that all changes are on target.

Whilst off target mutations arising from CRISPR-Cas9 mediated gene editing are a widely reported phenomenon, in the context of the generation of *in vitro* derived red blood cells, the importance of these non-specific mutations is mitigated by the stringent selection imparted by the terminal differentiation and enucleation processes for the generation of a functional reticulocyte. Nevertheless, in order to assess the frequency and identity of off target mutations within this clonal 5x KO BEL-A line and to satisfy the reviewers concerns regarding off target effects of the editing process we have performed whole genome sequencing of the unedited and 5x KO BEL-A cell lines. No somatic SNPs or indels unique to the CRISPR-Cas9 edited 5x KO cell line were identified within the vicinity of sequences homologous (tolerance <6bp mismatch to guide sequence or where 10bp sequence upstream of PAM is identical) to the guides used illustrating the absence of off target mutations resulting from the editing process. Serological confirmation of null phenotypes (Figure 2C) and quantitative comparative proteomics (Figure 2E, Table 2, Extended View Figure 4) are provided in the manuscript, furthering the characterisation of the reticulocytes derived from the 5x KO line.

Additionally, performing such genome editing experiments in primary HSCs would be valuable to demonstrate that this can be done in a primary cell setting that may more effectively be translated to the clinical setting.

Regarding performing the editing experiments in primary HSCs, the nature of CRISPR-mediated gene editing dictates that following transduction of primary cells, the genetic basis of any observed blood group null phenotype will be different for every cell within the non-clonal population and is at odds with the reviewers' desire for additional molecular characterisation of the edited cells. Any characterisation of reticulocytes derived from edited primary cells would need to be repeated for every experiment or culture, with potentially different results each time rendering characterisation moot. In contrast, modified BEL-A derived reticulocytes are genotypically clonal and can be differentiated to generate reticulocytes for characterisation whilst maintaining a sustainable source of edited cells (through continued expansion or as frozen stocks). We therefore respectfully argue that performing gene editing experiments in primary HSCs would not enhance the manuscript.

Referee #2 (Comments on Novelty/Model System for Author):

The use of the immortalized BEL-A cell line for downstream modification of erythrocyte membrane proteins is very exciting. It opens up the opportunity to create tailor-made erythrocytes for transfusion to immunized patients for whom finding compatible blood is a major challenge. Whether this approach of multiple knockout will succeed in producing viable, functional erythrocytes remains to be seen however the potential cannot be ignored.

Referee #2 (Remarks for Author):

General Comments

This is a concise and exciting proof-of-principle paper to demonstrate the feasibility of using CRISPR-mediated gene editing to knock-out multiple integral erythrocyte membrane proteins without apparent gross structural damage. The premise, i.e. creation of a "universal" donor erythrocyte, is a goal for many research group in the field, and these investigators demonstrate that it can be done, at least technically. There is so much more to be done to determine how viable these cells are in the long term, but as all work in this exciting field, the knowledge gained into the biology of is as important as achieving the end goal of ex vivo erythrocyte production for use in transfusion therapy.

The first, and perhaps major, criticism is that the title of the paper does not reflect the content. There is no evidence given to show "Enhancement of red blood cell transfusion compatibility". The authors have not demonstrated by simple haemagglutination that the 5x KO cells they have generated are indeed compatible with the plasma of at least one of the 48/56 patients for which they claim these cells would be a better transfusion alternative. I recommend either that the title is reworded or, more clinically relevant, that the cell lines are tested with representative plasma samples to give credit to the current title.

We thank the reviewer for this suggestion and agree that illustrating this would enhance the manuscript. Plasma samples from the surveyed patients were not available therefore we have addressed the review's concern by conducting serological tests of the 5x KO cells using human sera samples curated by the International Blood Group Reference Laboratory containing alloantibodies to each of the ablated blood groups. 5x KO cells did not react to any of the antisera tested and in all cases passed through the gel and pelleted at the bottom of the microtubes. In contrast, untransduced control BEL-A cells agglutinated upon exposure to antibodies and remained trapped in the gel. The gel card method is used routinely in clinical settings for blood group typing, we thus now present evidence that these cells are compatible with immunoreactive sera and provide justification for the title.

It was interesting to read on page 4 and see in Figure EV1 that expression of CD47 was

considerably reduced and it will be interesting to see the long term effect of this change on cell survival.

Indeed, the reduced CD47 observed on the 5x KO fits with the previously published levels on Rh_{null} erythrocytes. This observation is backed up by the proteomics analysis which also shows reduced CD47 expression in 5x KO cells. The fact that CD47 is not reduced on any of the individual edits in which GPB, Duffy, Kell or the H antigen are removed is further in keeping with the hypothesis that the removal of RhAG and Rh is responsible for CD47 reduction through destabilisation of the Rh subcomplex. In addition, proteomic analysis also showed major reduction in the level of another Rh subcomplex protein, ICAM4. The role of CD47 in maintaining circulating RBC survival is a continuing source of debate within the red blood cell field. Although a prevailing dogma maintains that levels of CD47 decrease during circulatory ageing and that this mediates clearance, evidence in support of this is mixed. Importantly, the severely reduced levels of CD47 found on Rh_{null} erythrocytes was not found to be associated with enhanced phagocytosis by circulating monocytes (Arndt & Garratty, 2004), the effective density of protein required to exert an anti-phagocytic effect was also reported to be lower than that observed in erythrocytes with CD47 deficiency due to absence of Rh (Tsai & Discher, 2008) and erythrocytes with D. and D—. Rh variants with similar levels of CD47 to that of Rh_{null} (and 5x KO cells reported here) show no signs of anaemia or enhanced clearance (Mouro-Chanteloup et al, 2003). These studies suggest that a reduction of CD47 to the level observed is unlikely to result in premature clearance of these cells *in vivo*.

Specific comments (note: the manuscript was not paginated and thus numbers below are assigned from the title page forward. Line numbers are given including paragraph titles, starting with Abstract)

Page 1, Line 21

There are currently 352 blood group antigens recognised by the ISBT, not "more than 400".

Personal correspondence with an ISBT Working Party member informed us of two recent additions to the ISBT recognised blood group antigens bringing the current total up to 354. The ISBT reference tables are yet to be updated therefore we have corrected the manuscript to "more than 350" antigens.

Page 2, lines 12-14

This sentence states that "Current efforts to cater..." then lists a reference from 1999. There are more recent references that would make this sentence more credible. Two are given here: S, Scharberg EA, Thornton N, Yahalom V, Sareneva I, Lomas-Francis C. International rare donor panels: a review. Vox Sang. 2016 Apr;110(3):209-18. Nance ST. How to find, recruit and maintain rare blood donors. Curr Opin Hematol. 2009 Nov;16(6):503-8.

Adjusted as requested

Page 7, line 3

It appears that there is an extra "+1%BSA" in this sentence describing the reagent PBSAG.

Adjusted as requested

Page 11, Table 1

It was interesting to note that antigens such as C^W, K, Kp^a, and Lu^a are listed as difficult to match since these are of low prevalence. Also, the use of ISBT nomenclature is recommended for describing blood group antigens, i.e. Kp^a not Kpa, etc.

Table 1 provides a summary of alloantibodies that were detected within plasma from the 56 patients identified with rare or problematic transfusion requirements over the given time period, therefore the antigens listed rather than referring to a list of antigens within the respective blood systems that may require theoretical matching in fact refer to actual alloantibodies identified in at least one of the patients.

Page 13, Figure 1

The low signal given by the untransduced BEL-A cells when tested with BRIC 203 was surprising. Is there an explanation for the weak expression of the Kell glycoprotein in these cells?

In the original figure, fixed cells were used for BRIC 203 labelling. Fixation of reticulocytes appears to reduce the intensity of BRIC 203 labelling. We have repeated the labelling using unfixed cells which results in an increased labelling intensity for the untransduced BEL-A derived reticulocytes. The signal from BRIC 203 labelling of the unfixed 5x KO cells remains completely absent. In order to provide additional confirmation of the Kell null phenotype, an alternative monoclonal antibody BRIC 68 was also used to confirm the null phenotype (figure below) and the absence of Kell is further supported by the serological data we present as a new Figure 2C.

Page 14, Figure 2

I recommend that the upper and lower panels in A are switched to reflect the initial knockout of GPB prior to the subsequent knockout of Duffy, RhAG, FUT and Kell.

We agree with the reviewer and have adjusted this as requested.

In panel C, it would be simpler to show the Rheoscope results of the two 5x KO cells individually rather than giving standard deviation, which is not relevant in such a small sample size.

Since the initial submission we have performed additional repeats of ARCA analysis on independent cultures of 5x KO and now present the data as the mean of 5 independent cultures with accompanying standard deviation.

Table EV2

Please list phenotypes according to ISBT nomenclature.

Adjusted as requested

Referee #3 (Comments on Novelty/Model System for Author):

The manuscript lacks any molecular analysis of the resulting cells with only one assay (deformability) to test the physical properties of the resulting cell line.

Referee #3 (Remarks for Author):

This is a very nice proof of principle paper where the authors use their recently developed BEL-A cell line to show that at least five red blood cell antigen can be knocked out without substantially changing the capacity of the cells to differentiate to enucleated RBCs and that their deformability index has hardly changed. The ability to do this is a step forward to obtain RBCs for transfusion in (rare) patients that develop an immune reaction to these RBC antigens. The method to derive these cells was a sequential KO using Crispr/Cas with different guide RNAs for the different genes. These were delivered via viral infection and stable cell lines were obtained that the authors show are convincingly negative for the targeted antigens. They subsequently show that the deformability of the RBCs obtained after differentiation of these cells is very comparable to that of wt cells. It is disappointing that the authors do not provide any molecular analysis of their KO cells. It is unknown what mutations actually occurred, whether there are bystander effects (particularly for 5

KO's). Best would have been of course to sequence the genome of the cells before and after the KO's to obtain a precise picture of the process. After all this is presented as a way to get to "universal" donor cells. A number of groups are pursuing similar or very different methods to achieve a similar goal and hence it is important to carefully describe the results even if it is only presented as a proof of principle. Thus I recommend that the authors revise the manuscript both at the level of molecular and physical property analysis.

We thank the reviewer for their positive comments regarding the manuscript and have taken on board suggestions to provide further characterisation. In order to provide additional molecular analysis of the KO cells as requested we have conducted whole genome sequencing analysis of reticulocytes derived from both 5x KO and untransduced control lines, characterising the individual gene edits resulting in KO of the five genes and providing evidence of absent off-target editing effects (see also response to Reviewer 1). Further characterisation of the reticulocytes is provided through comparative quantitative proteomic analysis of 5x KO reticulocytes. We have included an additional Figure 2E in the manuscript showing the change of membrane and cytoskeletal protein abundance and a Table 2 listing the fold change in abundance of important membrane and cytoskeletal proteins for red blood cell structure and function. Only 0.02% of membrane and cytoskeletal proteins showed at least a two-fold change in abundance demonstrating minimal structural disruption. The full proteomic dataset is also provided in the supplemental information. In addition, serological testing of the 5x KO (Figure 2C) confirmed the null phenotypes as these cells did not agglutinate when exposed to antibodies in human antisera providing a clinically relevant characterisation.

References

Arndt PA, Garratty G (2004) Rh(null) red blood cells with reduced CD47 do not show increased interactions with peripheral blood monocytes. *British journal of haematology* **125**: 412-414

Kim YH, Kim HO, Baek EJ, Kurita R, Cha HJ, Nakamura Y, Kim H (2015) Rh D blood group conversion using transcription activator-like effector nucleases. *Nature communications* **6**: 7451

Mouro-Chanteloup I, Delaunay J, Gane P, Nicolas V, Johansen M, Brown EJ, Peters LL, Van Kim CL, Cartron JP, Colin Y (2003) Evidence that the red cell skeleton protein 4.2 interacts with the Rh membrane complex member CD47. *Blood* **101**: 338-344

Trakarnsanga K, Griffiths RE, Wilson MC, Blair A, Satchwell TJ, Meinders M, Cogan N, Kupzig S, Kurita R, Nakamura Y, Toye AM, Anstee DJ, Frayne J (2017) An immortalized adult human erythroid line facilitates sustainable and scalable generation of functional red cells. *Nature communications* **8**: 14750

Tsai RK, Discher DE (2008) Inhibition of "self" engulfment through deactivation of myosin-II at the phagocytic synapse between human cells. *The Journal of cell biology* **180**: 989-1003

3rd Editorial Decision

8 March 2018

Thank you for the submission of your revised manuscript to EMBO Molecular Medicine. We have now received the enclosed reports from the referees that were asked to re-assess it. As you will see the reviewers are now globally supportive and I am pleased to inform you that we will be able to accept your manuscript pending the following final amendments:

1) Globally supportive but not fully: you will see that referee 1 remains unsatisfied. Could you please reply to this referee's comments in writing in a point-by-point response?

Referee #1 (Remarks for Author):

I have carefully reviewed this revised manuscript from Hawksworth and colleagues. The authors have argued against many of the criticisms and concerns I had raised in my initial review. Unfortunately, my concern remains that this is a study of limited general value in a single cell line that has not been well characterized. I can only find a single paper that uses this transformed BEL-A

cell line (from many of the same authors as this paper). The value for transfusion medicine of such a transformed cell line is unclear. The whole genome sequencing analysis performed by the authors is commendable, but beyond this, only superficial modifications have been made. I believe that this work is more appropriate for a specialized journal. The general readership of EMBO Molecular Medicine is unlikely to recognize the significant limitations of the work as presented.

Referee #2 (Comments on Novelty/Model System for Author):

As I wrote previously, the use of the immortalized BEL-A cell line for downstream modification of erythrocyte membrane proteins is very exciting. It opens up the opportunity to create tailor-made erythrocytes for transfusion to immunized patients for whom finding compatible blood is a major challenge. The improvements to the manuscript included in this revision strengthen the data to support the validity of this approach.

Referee #2 (Remarks for Author):

The authors have adequately addressed previous remarks. The manuscript is greatly improved by the visual evidence of enhanced compatibility as shown by the gelcard hemagglutination testing. The addition of the proteomics data also strengthens the authors' premise that these cells are otherwise functional.

Referee #3 (Comments on Novelty/Model System for Author):

The authors have appropriately addressed the comments and I recommend acceptance.

2nd Revision - authors' response

22 March 2018

We thank Reviewer 2 and 3 for their supportive comments and recommendation of acceptance.

We note that during the preparation of the final revision, a calculation error in the initial response to reviewers was detected "Only 0.02% of membrane and cytoskeletal proteins showed at least a two-fold change in abundance demonstrating minimal structural disruption" should read "2% of membrane and cytoskeletal proteins showed at least a two-fold reduction in abundance demonstrating minimal structural disruption". We apologise for this error. This does not affect the conclusion of minimal structural disruption arising from this data.

In response to Reviewer 1:

The authors have argued against many of the criticisms and concerns I had raised in my initial review. Unfortunately, my concern remains that this is a study of limited general value in a single cell line that has not been well characterized. I can only find a single paper that uses this transformed BEL-A cell line (from many of the same authors as this paper).

We are disappointed by the reviewer's comment and disagree strongly with their position on this point. Far from 'not well characterised', the BEL-A cell line (only published in 2017) and most relevantly the reticulocytes derived from it have been extensively characterised both in their unedited state and where edited for the generation of a five-blood group knock out RBC. Given the inclusion here of the whole genome sequence and entire reticulocyte proteome in addition to serological characterisation and deformability assessment it is difficult to understand how the reviewer is able to reach this assessment.

The value for transfusion medicine of such a transformed cell line is unclear.

As highlighted by Reviewers 2 and 3, production of *in vitro* derived red blood cells from sustainable cell sources including induced pluripotent stem cells and immortalised erythroblast lines is an exciting possibility currently being pursued by multiple groups using different approaches. Efforts towards the development of such cells are of clear interest to the transfusion medicine field however realisation of this potential for the production of a clinical transfusion product will require a variety of challenging technical obstacles to be overcome as clearly discussed in the manuscript.

We present here a proof of principle study that directly demonstrates the ability to use genome editing to ablate expression of multiple genes in immortalised erythroblasts to generate extensively characterised reticulocytes with modulated surface antigen profile.

In addition to highlighting the technical feasibility of this approach, this work demonstrates the important capacity for coexistence of multiple blood group null phenotypes within a viable red blood cell and reports both important methodology and a bank of lines that could be used as serological diagnostic tools.

The whole genome sequencing analysis performed by the authors is commendable, but beyond this, only superficial modifications have been made.

We are pleased that the reviewer recognises the inclusion of whole genome sequencing in the revised manuscript however the subsequent comment that ‘beyond this only superficial modifications have been made’ is inaccurate and at odds with the assessment of the other reviewers. For clarity, in addition to whole genome sequencing of the unedited and 5xKO lines, additional data includes quantitative comparative reticulocyte proteomics, serological confirmation of null phenotypes and additional independent replicates of deformability data. Each addresses comments or requests from the initial reviews, represents significant work and together most certainly address this reviewer’s requirement for ‘detailed characterisation of the cells’.

I believe that this work is more appropriate for a specialized journal. The general readership of EMBO Molecular Medicine is unlikely to recognize the significant limitations of the work as presented.

We strongly disagree with the reviewer on this comment which we believe underestimates the broad interest in this research area and the application of CRISPR/Cas9 and disservices the informed readership of EMBO Molecular Medicine.

Corresponding Author Name: Dr Ashley Toye

Manuscript Number: EMM-2017-08454-V3